# Implications of the Metabolic Control of Diabetes in Patients with Frailty Syndrome

**DOI:** 10.3390/ijerph191610327

**Published:** 2022-08-19

**Authors:** Marta Muszalik, Hubert Stępień, Grażyna Puto, Mateusz Cybulski, Donata Kurpas

**Affiliations:** 1Department of Geriatrics, Faculty of Health Sciences, Nicolaus Copernicus University in Torun, Collegium Medicum in Bydgoszcz, 85-094 Bydgoszcz, Poland; 2Department of Internal Medicine II, Asklepios Clinic Uckermark, 74-506 Schwedt, Germany; 3Department of Clinical Nursing, Institute of Nursing and Midwifery, Faculty of Health Sciences, Jagiellonian University Medical College Cracow, 31-501 Crakow, Poland; 4Department of Integrated Medical Care, Faculty of Health Sciences, Medical University of Bialystok, 15-096 Białystok, Poland; 5Department of Family Medicine, Wroclaw Medical University, 51-141 Wroclaw, Poland

**Keywords:** diabetes mellitus, frailty syndrome, metabolic control, older adults, functional capability, MMSE, ADL, IDAL, GDS, SHARE-FI

## Abstract

Introduction: Frailty syndrome occurs more frequently in patients with diabetes than in the general population. The reasons for this more frequent occurrence and the interdependence of the two conditions are not well understood. To date, there is no fully effective method for the diagnosis, prevention, and monitoring of frailty syndrome. This study aimed to assess the degree of metabolic control of diabetes in patients with frailty syndrome and to determine the impact of frailty on the course of diabetes using a retrospective analysis. Materials and Methods: A total of 103 individuals aged 60+ with diabetes were studied. The study population included 65 women (63.1%) and 38 men (36.9%). The mean age was 72.96 years (SD 7.55). The study was conducted in the practice of a general practitioner in Wielkopolska in 2018–2019. The research instrument was the authors’ original medical history questionnaire. The questions of the questionnaire were related to age, education, and sociodemographic situation of the respondents, as well as their dietary habits, health status, and use of stimulants. Other instruments used were: the Mini-Mental State Examination (MMSE), Lawton Scale (IADL—Instrumental Activities of Daily Living), Katz Scale (ADL—Activities of Daily Living), Geriatric Depression Rating Scale (GDS), and SHARE-FI scale (Survey of Health, Aging, and Retirement in Europe). Anthropometric and biochemical tests were performed. Results: In the study, frailty syndrome was diagnosed using the SHARE-FI scale in 26 individuals (25%): 32 (31.1%) were pre-frailty and 45 (43.7%) represented a non-frailty group. Statistical analysis revealed that elevated HbA1c levels were associated with a statistically significant risk of developing frailty syndrome (*p* = 0.048). In addition, the co-occurrence of diabetes and frailty syndrome was found to be a risk factor for loss of functional capacity or limitation in older adults (*p* = 0.00) and was associated with the risk of developing depression (*p* < 0.001) and cognitive impairment (*p* < 0.001). Conclusions: Concerning metabolic control of diabetes, higher HbA1c levels in the elderly are a predictive factor for the development of frailty syndrome. No statistical significance was found for the other parameters of metabolic control in diabetes. People with frailty syndrome scored significantly higher on the Geriatric Depression Rating Scale and lower on the MMSE cognitive rating scale than the comparison group. This suggests that frailty is a predictive factor for depression and cognitive impairment. Patients with frailty and diabetes have significantly lower scores on the Basic Activities of Daily Living Rating Scale and the Complex Activities of Daily Living Rating Scale, which are associated with loss or limitation of functioning. Frailty syndrome is a predictive factor for loss of functional capacity in the elderly.

## 1. Introduction

The population of older people (defined here as those aged 65 or more) in the EU-27 is projected to increase substantially, from 90.5 million at the beginning of 2019 to 129.8 million in 2050. During this period, the number of people aged 75–84 in the EU-27 is projected to increase by 56.1%, while the number of people aged 65–74 will increase by 16.6%. In contrast, according to the latest projections, there will be 13.5% fewer people under the age of 55 in the EU-27 by 2050 [1].

Aging is a long-term, irreversible physiological process that occurs at three main levels: biological, psychological, and social. The changes that occur in the body as a result of the aging process affect all systems and lead to a decline in performance and deterioration of physical and mental functions. One of the syndromes that significantly affects the course of the aging process is frailty syndrome.

Frailty syndrome is characterized by a decrease in physiological reserves and increased sensitivity to stress factors that affect the homeostasis of the human body (impairment of the homeostasis of the body). Therefore, the risk of loss of independence, hospitalization, institutionalization, and mortality increases in this group of patients. In 2001, Linda Fried et al. defined frailty syndrome as “a biological syndrome of diminished reserve and resilience to stressors resulting from cumulative deterioration of multiple physiological systems, involving vulnerability to adverse consequences” [2].

According to Fried et al., the predictors of fragility syndrome are age-related characteristics, which include: decrease in muscle strength, decrease in lean body mass, deterioration of balance, decrease in endurance, decreased ability to move, and limitation of physical activity. To make a diagnosis of frailty syndrome, several of the above features must be present [2].

In addition to the clinical features mentioned above, other authors mention mood, cognitive function, nutritional status, strength, activity, endurance, and mobility, among others [3]. Santos-Eggimann et al. considered the Fried criteria in the first European study on the prevalence of frailty (Survey of Health, Aging, and Retirement in Europe (SHARE)) in a group of middle-aged (over 50) and community-dwelling older Europeans in ten countries. The researchers distinguished five SHARE criteria. It was the first attempt to operationalize Fried’s phenotype of frailty in a very large European population-based sample. The prevalence of frailty in European society (65 years and older) ranges from 5.8% to 27.3%, while 34.6% to 50.9% of older adults are considered ‘pre-frail’ [4]. The SHARE-FI questionnaire is an effective research tool for diagnosing frailty syndrome in European seniors [5,6,7,8]. Current definitions of the frailty complex emphasize the importance of the multidimensional concept and the integral approach to human functioning [9]. According to Gilardi et al., a multidimensional approach to frailty is more effective for planning and implementing care services and for planning prevention programs for the frail elderly [10].

Frailty is a dynamic condition in which physical, psychological, and social factors interact to disrupt the homeostasis of the organism and lead to negative consequences [11]. Research shows that the risk of developing frailty is related to the female gender, low educational level, and income, and increases with age, development of chronic diseases, and disability [12].

Diabetes is recognized by the World Health Organization—WHO—as one of the most important social threats and is the only non-communicable disease classified as an epidemic. In 2017, 123 million people over the age of 65 had diabetes worldwide, and this number is expected to double by 2045. Elderly diabetics are at higher risk of developing geriatric syndromes, including cognitive impairment and dementia, incontinence, falls, disability, polypharmacy, and frailty syndromes, which have a significant impact on a patient’s quality of life and the effectiveness of antidiabetic treatment [13]. At the functional level, older people with diabetes show a greater decline in functional capacity than healthy people of the same age. The development of diabetes is associated with a higher risk of disability, need for long-term care, and mortality [14].

With age, insulin resistance increases, and pancreatic cell function deteriorates. Normally, the body maintains homeostasis without problems, but in obesity or genetic stress, glucose tolerance may be impaired. Greater insulin resistance is observed in older people than in younger people. However, it is controversial whether this is a consequence of the biological aging of the organism or environmental and lifestyle influences. In the elderly, it is observed that a reduction in fat-free mass, poor dietary habits, and lack of physical activity decrease insulin sensitivity [15,16,17]. It seems that insulin resistance in the elderly is mainly related to the loss of skeletal muscle, but not to the overproduction of glucose in the liver. Coexisting diseases and medication use may also contribute to the development of insulin resistance [18].

According to the latest estimates from the International Diabetes Federation (IDF), diabetes shows a high prevalence in people older than 65 years [8]. In 2017, the number of diabetics aged 65–99 years was estimated to be 122.8 million (about 18% of the prevalence rate), of whom 98 million were <80 years old (65–79 years); these numbers are expected to easily exceed 200 million in 2045 [19]. Several data suggest that diabetes is associated with frailty and disability in older adults. Diabetics are more likely to be frail than older adults without diabetes [20,21,22]. Early diagnosis of frailty in older diabetics allows for comprehensive, multifaceted interventions, including exercise, nutritional support, medication adjustment, and disease control [23].

Management of diabetes requires medical, functional, psychological, and social assessment of the patient. In addition to cardiovascular and microvascular disease, older people with diabetes and frailty should be assessed for typical geriatric diseases and conditions such as cognitive impairment, risk of falls, depression, visual and hearing impairment, incontinence, eating problems, and others. A comprehensive geriatric assessment is a basis for setting individualized treatment goals. Treatment strategies for people with nonrefractory diabetes are like those for other adults. In the case of frailty syndrome, individualized care plans should be established according to functional status and life expectancy to minimize the risk of complications [24].

The therapeutic goals of diabetes treatment should be understood as achieving target values in the following areas: glycemia, lipid profile, blood pressure, and body weight. In the elderly with multiple diseases whose life expectancy is less than 10 years, treatment goals should be relaxed to the extent that the patient’s quality of life is not compromised. In general, these days goals and therapy should be individualized. The American Geriatric Society recommends a target glycosylated hemoglobin (HbA1c) of 7.5–8.0% for the elderly. However, glycemic control targets may vary depending on the patient’s condition. The HbA1c target of 7.0–7.5% is appropriate for the functionally independent elderly with reasonable life expectancy, whereas the target of 8–9% is appropriate for elderly patients in poor health and those with dementia and a life expectancy of fewer than 10 years [25].

Frailty and cognitive impairment are common in patients with metabolic and cardiac diseases, and diabetes increases this risk (neurodegenerative dementia) [26]. Patients with frailty are at increased risk for hypoglycemia because most of them lose their appetite and weight. Acute hypoglycemia causes neurological symptoms such as dizziness or visual disturbances and leads to deterioration of general condition and hospitalization. In the elderly, the risk of hypoglycemia and its consequences should be minimized. Therefore, the prevention of hypoglycemia should always be a priority in the management of elderly patients with diabetes and frailty [27]. According to studies, frailty is the main factor that increases the risk of death and disability in older people with diabetes [28]. Frailty is also associated with insulin resistance in the post-absorptive state of glucose metabolism when more abdominal fat is present [29].

To date, there is no fully effective method to detect, prevent, and monitor vulnerability syndrome. Assuming that the diagnosis of vulnerability syndrome is a factor that worsens the metabolic control of diabetes, it is possible to diagnose it early and treat it effectively.

Our study is comprehensive and multi-stage, with the use of geriatric assessment elements, many research tools, as well as physical and biochemical tests. It is also retrospective and comparative, which is not found in other studies. Such a multi-faceted approach broadens the image and knowledge of the functioning of patients with diabetes and frailty. Each patient diagnosed with frailty was subjected to further detailed examinations. Each patient was provided education on further dietary management and physical exercises individually. The latest knowledge in this field guided the doctor.

This study aimed to evaluate the degree of metabolic control of diabetes in patients with frailty syndrome and to determine the influence of frailty syndrome on the course of diabetes using a retrospective analysis.

## 2. Materials and Methods

### 2.1. Data Collection

Studies were conducted from January 2018 to March 2019 in 103 outpatients with type 2 diabetes aged 60+ (mean age 72.96, SD = 7.55). Patients with hemiparesis, oncological diseases, major depression or other mental disorders, and severe cognitive impairment were excluded from the study. The study was conducted individually with each participant. The patient gave informed and written consent to the study. Each participant, before agreeing to participate in the study, was informed about the aims of the study, how to participate, the anonymity of the study, and the possibility of withdrawing from participation at any time. The study was conducted using a questionnaire administered by a physician and a qualified nurse (a member of the research team).

### 2.2. Ethical Procedures

The research was conducted from February 2018 to March 2019 and approved by the Bioethics Committee of Nicolaus Copernicus University in Torun, Collegium Medicum, in Bydgoszcz (KB 395/2017). The study was conducted following the principles of the Declaration of Helsinki.

### 2.3. Participants

A total of 103 patients with diabetes participated in the study. Most of them were farmers (43, 41.7%), housewives (21, 20.4%), or office workers (11, 10.7%). Patients mostly had two (37, 35.9%) or three (19, 18.4%) children. Most of the respondents were taking 11 medications, with a median of 5 medications.

The results of the examination of the cognitive and functional status of the entire group of respondents were in the upper range. Regarding emotional state, 50 subjects (48.54%) had no depression and 53 subjects (51.46%) had mild depression.

The characteristics of the study group are shown in Table 1.

### 2.4. Study Design

#### 2.4.1. Stages of the Study

In the first phase, a subjective examination was conducted, which included a subjective assessment of the participant’s health, the number of medications taken, complaints, limitations in the quality of life, consumption of stimulants (alcohol, cigarettes), and selected lifestyle elements (physical activity).

Then, the physical condition of the participants was examined. Anthropometric tests were performed: measurement of body weight and height, waist circumference, handgrip strength, a test of getting up on a chair, walking speed, and impact force.

In the second phase of the study, the biochemical blood parameters were measured: glucose, HbA1c, total cholesterol, triglycerides, LDL, and HDL. Then, arterial blood pressure and heart rate were studied. The occurrence of frailty syndrome was determined by the method of direct patient assessment using the SHARE-FI scale. Then, the test group and the control group were separated. The comparison group consisted of patients who had been diagnosed with diabetes but did not meet the criteria for a diagnosis of frailty syndrome.

In the third phase of the study, a retrospective and comparative analysis of the medical data (HbA1c, lipid profile, anthropometric data, systolic and diastolic blood pressure) of the patients included in the study was performed.

Subsequently, the content of the collected research material was reviewed, and statistical analysis was performed. An original questionnaire was used in the study, which contained open and closed questions with the possibility of single or multiple choices. The questions in the questionnaire were related to age, education, socio-demographic data of the respondents, eating habits, physical activity, health status, and consumption of stimulants (alcohol, cigarettes).

#### 2.4.2. Research Tools

The following scales were used in the study: Mini-Mental State Examination (MMSE), Lawton Scale (IADL—Instrumental Activities of Daily Living), Katz Scale (ADL—Activities of Daily Living), Geriatric Depression Scale, and the scale SHARE-FI (Survey of Health, Aging, and Retirement in Europe).

##### Mini-Mental State Examination (MMSE)

Cognitive performance was evaluated using the Polish version of the Mini-Mental State Examination (MMSE) scale. The scale consists of 30 questions which allow for the quantitative assessment of different aspects of cognitive functioning. The areas subject to evaluation are as follows: orientation to time and place, registration, attention and calculation, recall, language, repetition, reading comprehension, writing, and drawing. The maximum score is 30 points. The cut-off point of 23 is a sensitive indicator of cognitive decline and indicates the need for specialized testing. The obtained score can also be related to a specific category: 27–30 is a normal result, 24–26 signifies cognitive disorders without dementia, 19–23 suggests mild dementia, 11–18 denotes moderate dementia, and 0–10 reflects deep dementia [30].

##### Geriatric Depression Scale (GDS)

Depression was assessed using a 15-point Geriatric Depression Scale (GDS). This is a screening tool that is used to evaluate the severity of depression symptoms in older people. The interpretation of the results is as follows: 0 to 5 points mean healthy condition, 6 to 10 points signify a moderate sense of depression, and 11 to 15 points signify a deep sense of depression [31].

The assessment of a range of Complex Activities of Daily Living used the Instrumental Activities of Daily Living (IADL) scale. The IADL scale assesses instrumental activities in eight areas of functioning, i.e., ability to use the telephone, housekeeping, shopping, food preparation, mode of transportation, ability to handle finances, and responsibility for own medications. The total score is relevant to a particular patient and a fall in the score on consecutive examinations reflects a deteriorated general state. The result of the activity of daily living assessment of an older person on that scale allows for objectivizing the patient’s needs for care or necessary assistance [32].

The Activities of Daily Living (ADL) scale is the most appropriate instrument to assess functional status as a measurement of the patient’s ability to perform activities of daily living independently. This scale measures six functions: bathing, dressing, toileting, transferring, continence, and feeding. Patients are scored yes/no for independence in each of the six functions. A score of 6 indicates full function, 4 indicates moderate impairment, and 2 or less indicates severe functional impairment [33,34].

Frailty syndrome was assessed using the Survey of Health, Aging, and Retirement in Europe SHARE-FI, which is recommended as a screening test in primary health care and hospital care in people over 60 years of age of both sexes [5]. The translation and validation procedure of the Polish version of the SHARE-FI was completed by M. Muszalik et al. The results of Cronbach’s alpha reliability coefficients of the SHARE-FI instrument ranged from 0.73 to 0.83, and the corrected item-total correlation ranged from 0.11 to 0.91 [35]. The questions included in the SHARE-FI concern the following areas: the subject’s gender, subjective feeling of exhaustion, loss of appetite, difficulty in walking upstairs, limitation in physical activity, and assessment of handgrip strength (measured with a manual hydraulic dynamometer SAEHAN SH501). The obtained numerical values calculated with the use of the SHARE-FI virtual calculator allow the subject to be classified into one of the three groups: non-frail, pre-frail, and frail [36]. Frail classification of the subject was frail: >3 for men and >2.13 for women; pre-frail: 1.21–3 for men and 0.32–2.13 for women; non-frail: <1.21 for men and <0.32 for women [4].

### 2.5. Statistical Analysis

The program IBM SPSS Statistics 25 was used for statistical analysis. The significance level was set at *p* ≤ 0.05. Shapiro–Wilk test was used to test the normality of data distribution and Levene’s test was used to test the homogeneity of variance. The data were compared using the Mann–Whitney U test. First, all data were tested for distribution using the Shapiro–Wilk test, separately for individuals with and without frailty syndrome. Much of the data were normally distributed (value above the significance level), but in some cases, the *p*-value was below the significance level. In addition, Levene’s test showed a lack of homogeneity of variance in many places. Moreover, the studied groups differed significantly in the number of patients (the group with the frailty syndrome—26 people, the group without the frailty syndrome—77 people). Therefore, a group of non-parametric tests was selected to compare the data distributions. For all variables, the mean and standard deviation were recalculated, taking into account the division into groups concerning the presence of the frailty syndrome.

## 3. Results

### 3.1. Main Findings Regarding Objective Measures of the Subjects’ Physical Health

Objective measures of physical health were tested: handgrip, air pressure, walking speed, and chair stance test. The results show that the average hand pressure strength was significantly higher in the studied population compared to existing studies [35]. This is most likely related to the characteristics of the studied population, in which 41.7% are farmers, who had significantly better results compared to the rest of the population. As for the other parameters, the results did not differ from the available data.

The results are presented in Table 2.

### 3.2. Main Findings Regarding the Comparison of the Frail and Non-Frail Groups

Finally, frailty syndrome was diagnosed in 26 subjects (25.2%), 32 subjects (31.1%) were diagnosed with the condition of pre-frailty, and 45 subjects (43.7%) were without the condition of frailty.

Both groups of diabetics—with and without frailty syndrome—were compared with each other using the non-parametric Mann–Whitney U test concerning all analyzed values.

People with frailty syndrome had significantly lower current weight (*p* <0.001) and lower weight 3 years ago (*p* < 0.001), and consequently significantly lower current BMI value (*p* = 0.003) and lower BMI 3 years ago (*p* = 0.006). Higher HbA1c levels are associated with the risk of developing frailty syndrome in subjects (*p* = 0.048). Measures of right and left hand grip strength were significantly lower in the frail group than in the group without frailty (*p* < 0.001).

On the GDS scale, people with frailty syndrome had a significantly higher risk of developing depression (*p* < 0.001). On the MMSE scale, people with frailty syndrome had lower scores than non-frail individuals (*p* = 0.011). The non-frail patients obtained significantly better results in the chair rising test (*p* < 0.001).

Detailed results can be found in Table 3 and Figure 1, Figure 2 and Figure 3.

The figures show the mean values obtained by the participants of the study. 

## 4. Discussion

The study aimed to evaluate the degree of metabolic control of diabetes in patients with frailty syndrome and to determine the influence of frailty syndrome on the course of diabetes using a retrospective analysis.

In the preliminary studies, the study group and the control group were divided. The study group consisted of patients with diagnosed diabetes and newly diagnosed frailty syndrome. The comparison group consisted of patients diagnosed with diabetes but who did not meet the criteria for frailty syndrome. Frailty syndrome was assessed by the direct patient assessment method using the SHARE-FI scale. Recent scientific reports show that the Polish version of SHARE-FI has high coefficients for internal consistency and reliability and can be recommended for screening people aged 60+ in primary care and hospital settings [35]. In both groups, the degree of metabolic control of diabetes was assessed by biochemical, physical, and anthropometric parameters that, according to the available scientific data, potentially influence the degree of metabolic control of diabetes. Their incorrect level is associated with a high risk of complications. Analysis of available studies shows that frailty syndrome is considered a risk factor for cardiovascular and metabolic diseases. In elderly people with frailty syndrome, morbidity and mortality due to cardiovascular diseases are statistically higher than in the rest of the population [37].

According to the studies performed so far, frailty and cardiovascular disease share a common pathogenesis, as both conditions are associated with low-grade chronic inflammation. Chronic inflammation in cardiovascular disease leads to oxidation of lipoproteins and activation of atherosclerotic plaques. The inflammatory process in people with frailty syndrome contributes to the development of catabolic states, particularly in skeletal muscle, by redistributing amino acids from muscle. These changes also affect other organs and systems. Currently, the negative effects of frailty syndrome on musculoskeletal, endocrine, circulatory, hematopoietic, and nutritional status have been documented. Consequently, there is a strong association between frailty syndrome and chronic inflammation [38,39]. In our study, frailty syndrome was found in 26 subjects (25.2%), pre-fragile status in 32 subjects (31.1%), and 45 subjects (43.7%) with robustness. As a result of a statistical analysis of 103 patients over 60 years of age with type 2 diabetes, it was found that people with frailty syndrome had a significantly lower subjective assessment of their health (*p* = 0.001). In this group, there were significantly lower scores on the Katz Scale for Basic Activities of Daily Living (*p* = 0.00) and lower scores on the Lawton Scale for assessing Complex Activities of Daily Living (*p* < 0.001). Based on our research, it can be concluded that frailty syndrome is a factor that worsens functional performance in older people with diabetes. Analysis of the literature confirms that diabetes increases the risk of motor disability and disability in IADL and ADL. Wong et al. pointed out in their literature review that diabetes increases the risk of mobility disability (15 studies; OR 1.71, 95% CI 1.53–1.91; RR 1.51, 95% CI 1.38–1.64), IADL disability (ten studies; OR 1.65, 95% CI 1.55–1.74), and ADL disability (16 studies; OR 1.82, 95% CI 1.63–2.04; RR 1.82, 95% CI 1.40–2.36) [14]. Given the prevalence of diabetes worldwide, it is important to examine these aspects of people’s lives in the context of the global diabetes epidemic. Frailty syndrome and depressive symptoms are common problems in the elderly. It is unclear whether depression predisposes to the syndrome, occurs inversely, or exists independently. The coexistence of frailty syndrome and depression in people aged 65 years and older has only recently been studied [40,41,42]. Most of the published studies showed that people over 60 years of age with diagnosed frailty syndrome had increased depressive symptoms. Based on the Dutch Cohort Study of Depression in the Elderly (NESDO), the incidence of frailty syndrome was found to be three times higher in people with depression than in people without depression [43]. The prevalence rates of physical frailty were 27.2% and 9.1% in depressed and nondepressed participants, respectively, which remained significant after controlling for relevant covariates (odds ratio (OR) = 2.66 (95% confidence interval [C.I.] = 1.36, 5.24), *p* = 0.004). Physical frailty was associated with more severe depressive symptoms in depression [43].

The results of a study among elderly people in Mexico City showed that older age, disability, comorbidity, cognitive impairment, and depression could have an impact on frailty. According to the available studies, frailty and depression were also associated with an increased risk of developing dementia [44]. In a large Italian-population-based sample, frailty syndrome was a short-term predictor of dementia and VaD (vascular dementia) overall [45]. Our own study conducted indirectly confirmed the above hypothesis, as, on the MMSE cognitive state assessment scale, individuals with frailty syndrome had lower scores than those who did not suffer from this condition (*p* = 0.011). The physical condition was also assessed as part of the qualification process for the study. Statistical analysis revealed that people with frailty syndrome had statistically significantly lower parameters in handgrip strength (*p* < 0.001), bubble strength (*p* < 0.001), and stool holding test (*p* < 0.001) than the control group. The coefficient of walking speed (*p* = 0.005) was significantly higher in people with the weakness syndrome than in the control group.

The hand pressure strength parameter is one of the main criteria according to Fried. It is also considered a biomarker of aging and a predictor of disability, morbidity, and mortality. Handgrip strength was a better predictor of all-cause mortality and cardiovascular mortality than systolic blood pressure in more than 140,000 PURE participants aged 35–70 years PURE [46]. The pathogenesis of loss of upper limb compressive strength and decrease in gait velocity is closely related to the pathogenesis of the frailty syndrome described in the introduction.

Peak Expiratory Flow as a measure of respiratory function (PEF) is a parameter associated with adverse health outcomes in old age, such as disability and mortality. It has lower diagnostic accuracy compared with spirometry but may be an important screening alternative for older people who cannot undergo accurate spirometry. To some degree, a decline in PEF is a normal physiologic adaptation to aging. However, the above-average decline may be associated with pathological conditions affecting the cardiovascular and pulmonary systems, efficiency and expiratory function, and cognitive and physical performance [47]. Analysis of the authors’ research suggests that low PEF, lower hand press strength parameters, and decreased walking speed (*p* = 0.005) may be both determinants and predictors of frailty syndrome.

These simple tests can serve as screening tests for syndrome assessment. The relationship between PEF and frailty syndrome could be based on several mechanisms. Among them, the process of sarcopenia plays a key role. Sarcopenia—the loss of muscle strength, quantity, and quality—is characteristic of the aging process. This process affects various muscles, including respiratory muscles. Sarcopenia is key to most of the criteria proposed by Fried et al. to operationalize the fragility syndrome. Sarcopenia multifactorial impairs the respiratory function of the body, decreases the compliance of the chest wall, causes weight loss of the respiratory muscles, and impairs their functions. Through these processes, sarcopenia may partially explain the reduction in Peak Expiratory Flow Rate (PEF) in individuals with frailty syndrome [48].

Impaired expiratory and inspiratory function may impair the cough reflex and promote the occurrence of respiratory infections. Therefore, low PEF may indirectly increase susceptibility to respiratory infections, which have a strong impact on overall health status and the development of frailty. Finally, cardiorespiratory fitness may influence cognitive performance, and low PEF has already been associated with poorer cognitive function and a higher risk of dementia. The pathophysiological mechanisms of this association are still unclear, but chronic states of hypercapnia or hypoxemia may play a role in this association and impair cognitive function [49].

Regarding the impact of frailty syndrome on the metabolic control of diabetes, statistically significant changes were found in HbA1c levels (*p* = 0.048). The other parameters of the metabolic control of diabetes did not change significantly in both groups. From the study conducted, it can be concluded that an initially elevated HbA1c level is a risk factor for frailty syndrome. The pathomechanism for this association is complex. In other studies of patients with type 2 diabetes, blood glucose or HbA1c levels be associated with U-shaped disease risk, i.e., not only high but also low HbA1c levels have been associated with dementia, stroke risk, falls, and increased mortality [50].

In a retrospective cohort study of elderly people with type 2 diabetes, HbA1c level was associated with mortality. Mortality was significantly higher at an HbA1c level of 6% than at higher levels. However, the authors pointed out a limitation of the analysis performed and concluded that hypoglycemia and/or type of therapy in patients with type 2 diabetes could have a significant impact on the above analyses, which was not the subject of the study [16]. Schwartz et al. [51] investigated that the risk of falls in diabetic women (mean age 73.6 years) was four times higher in the insulin-treated group with HbA1c levels below 6% than in the group with levels above 8%. Another study suggests that only high blood glucose levels are associated with an increased risk of developing frailty syndrome compared with low levels [52]. Recently, however, other studies have shown a U-shaped association between blood glucose levels and frailty syndrome. Zaslavsky et al. [53] showed an association between low HbA1c levels and frailty in a group of elderly patients with type 2 diabetes. Pilotto et al. [54] also showed a U-shaped risk between HbA1c levels and frailty syndrome. The differences between these tests are related to the use of different criteria to assess frailty syndrome used to qualify for the test.

Higher blood glucose levels, and therefore higher HbA1c levels, may contribute to an increased risk of developing frailty syndrome because of several possible mechanisms. The most important factor could be insulin resistance or insulin deficiency associated with the progression of type 2 diabetes. Insulin normally has an anabolic effect on muscles, and when it is absent or resistance develops, the muscles atrophy. In addition, chronic hyperglycemia has an independent effect on microvascular and macrovascular complications, which can lead to impaired work of many organs and the development of debilitation syndrome. This mechanism is mediated by glucose, which causes cellular oxidative stress and chronic inflammation. This pathomechanism is also a factor in the development of frailty syndrome. Chronic hyperglycemia also causes mitochondrial dysfunction of skeletal muscle, which may partially explain the muscle weakness and poor muscle quality in elderly diabetics [55].

It has been shown that with increasing duration of diabetes and in patients with higher HbA1c levels, muscle weakness is more pronounced than in the rest of the population. Interestingly, higher HbA1c levels are associated with greater difficulty walking [56]. Kalyani et al. investigated that walking difficulty or performance impairment related to leg function was greater in individuals with an HbA1c level of ≥8% than in individuals with an HbA1c level of <5.5% at baseline. These scientific reports seem to be a reliable explanation for the association between hyperglycemia and weakness [56].

One of the causes that plays a key role in the development of the weakness syndrome is malnutrition.

From the study, it can be indirectly concluded that people with weakness syndrome had significantly lower body weight both currently and 3 years ago (*p* = 0.006), and consequently significantly lower BMI currently and 3 years ago (*p* = 0.003). These indicators are only indirect evidence because the study did not directly determine nutritional status. As we know, body weight in the elderly is influenced by other factors such as loss of muscle mass, chronic diseases, and others. Therefore, malnutrition in elderly patients with type 2 diabetes may be a risk factor for the development of frailty syndrome, as confirmed by the study of Wei et al. [57].

Malnutrition associated with type 2 diabetes may be the result of strict self-prescribed or physician-prescribed diets to control blood glucose levels. According to the available reports, eating disorders may pose a risk for the development of frailty syndrome in elderly patients with type 2 diabetes [58].

HbA1c levels in older people with diabetes cannot be explained simply by basic mechanisms. Regardless, in elderly diabetics, malnutrition or inadequate caloric intake in the diet seem to be the most likely factors for the association between frailty and low HbA1c levels. Malnutrition may be further modified or exacerbated by the presence of diabetic complications, hypoglycemic management, cognitive impairment, and disability. On the other hand, general disability caused by the syndrome may exacerbate malnutrition or inadequate nutrition in diabetic patients.

Thus, it is thought that lowering HbA1c and strict glycemic control may prevent the development of microvascular and macrovascular complications. Surprisingly, tight glycemic control did not reduce all-cause mortality in the Cardiovascular Risk Study (ACCORD) [59].

## 5. Limitations of the Study

The results of the above tests may change as further research is conducted. A type of change called reverse metabolism may be responsible. Reverse metabolism has been shown in several studies in people ≥ 85 years of age when hypertension, high cholesterol, and high blood glucose did not predict cardiovascular risk and mortality. Reverse metabolism syndrome requires further research. Most likely, malnutrition and/or chronic diseases play a key role in its development [60].

In our study, frailty syndrome correlated only with higher HbA1c levels. This difference could be due to the type of study method used and the size of the population studied. Nevertheless, tight glycemic control in some older diabetic patients may lead to poor prognosis in terms of mortality, disability, cardiovascular disease, cognitive impairment, and nutrition.

## 6. Conclusions

Regarding metabolic control in diabetes, higher HbA1c levels in the elderly are a predictive factor for the development of frailty syndrome. No statistical significance was found for the other parameters of metabolic control in diabetes. People with the syndrome scored significantly higher on the Geriatric Depression Rating Scale and lower on the MMSE cognitive ability rating scale than the comparison group. This suggests that frailty syndrome is a predictive factor for depression and cognitive impairment. Patients with frailty syndrome and diabetes have significantly lower scores on the Basic Activities of Daily Living Rating Scale and the Complex Activities of Daily Living Rating Scale, which are associated with loss or limitation of functioning. Frailty syndrome is a predictive factor for loss of functional capacity in the elderly.

## Figures and Tables

**Figure 1 ijerph-19-10327-f001:**
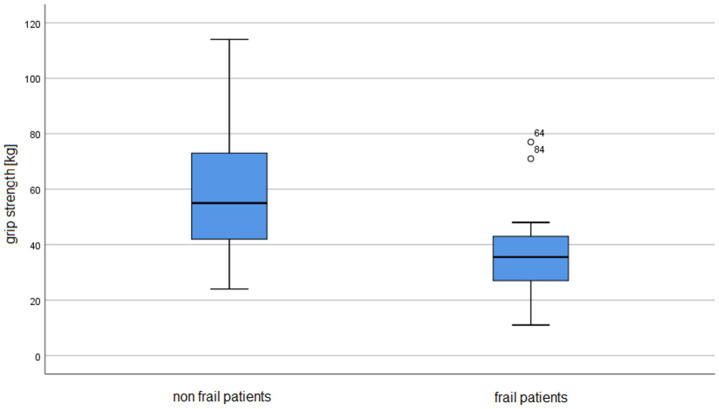
The result of the hand grip test in the study group.

**Figure 2 ijerph-19-10327-f002:**
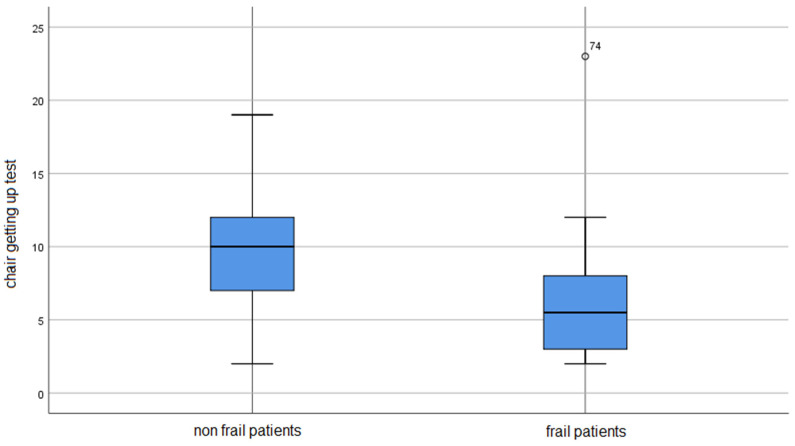
The result of the chair getting up test in the study group.

**Figure 3 ijerph-19-10327-f003:**
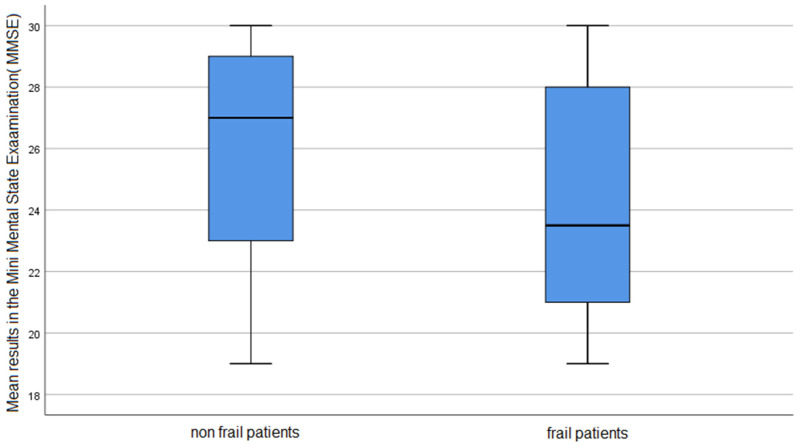
The result of MMSE in the study group.

**Table 1 ijerph-19-10327-t001:** Socio-demographic and clinical characteristics of the study group (n = 103).

Category	Mean/SD	n	%
Age	72.96/7.55	103	
Male	74.08/8.11	38	36.9
Female	73.71/7.78	65	63.1
Sex			
Male		38	36.9
Female		65	63.1
Marital status			
Married		65	63.1
Widow/widower/single		35	34.0/2.9
Indwelling			
Lonely		37	35.9
With family		66	64.1
Work			
Retirees		99	96.1
Pensioners		1	1.0
Working		3	2.9
Dwelling place			
City		26	25.2
Village		77	74.8
Health self-assessment			
Good		32	31.1
Medium		38	36.9
Badly		72	69.9
Social activity			
Active		10	9.7
Physical activity (yes)		16	15.5
Limitations on daily activities (yes)		63	61.2
Addictions			
Smoking (yes)		7	6.8
Drinking alcohol (yes)		9	8.7
Diabetic diet (yes)		64	62.1
MMSE 25.42/3.51			
ADL 5.7/0.6			
IADL 22/4.1			
GDS 8.8/4.5			

Abbreviations: n—number of respondents; %—a percentage of respondents; SD—standard deviation; MMSE—Mini-Mental State Examination; ADL—Activity of Daily Living; IADL—Instrumental Activity of Daily Living; GDS—Geriatric Depression Scale.

**Table 2 ijerph-19-10327-t002:** Objective measures of the subjects’ physical health (n = 103).

Physical Health Parameters	Grip Strength (kg)	Air Blowing Force (mL) PEF	Walking Speed( s)	Chair Getting up Test (the Number of Repetitions/30 s)
Mean	52.56	188.68	8.58	8.94
Medium	48.00	160.00	8.00	10.00
SD	22.234	105.524	3.713	4.454
Minimum	11	50	3	2
Maximum	114	600	24	23

Abbreviations: PEF—Peak Expiratory Flow Rate.

**Table 3 ijerph-19-10327-t003:** Comparison of the non-frail and frail groups using the Mann–Whitney non-parametric U test.

	U Test	Non-Frail/Frail N	*p*-Value
Health assessment	587.500	77/26	0.001
Grip strength	386.000	77/26	*p* < 0.001
Air blowing force	447.500	77/26	*p* < 0.001
Walking speed	1367.000	77/26	0.005
Chair getting up test	450.000	77/26	*p* < 0.001
Current growth	796.500	77/26	0.120
Growth 3 years ago	833.500	77/26	0.203
Current weight	485.000	77/26	*p* < 0.001
Weight 3 years ago	526.500	77/26	*p* < 0.001
Current waist circumference	766.000	77/26	0.074
Waist circumference 3 years ago	814.500	77/26	0.156
BMI currently	608.000	77/26	0.003
BMI 3 years ago	638.006	77/26	0.006
Heart rate currently	876.000	77/26	0.336
Heart rate 3 years ago	787.500	77/26	0.103
Glycemia currently	940.500	77/26	0.645
Glycemia 3 years ago	915.000	77/26	0.514
HbA1c currently	1550.500	77/26	0.241
HbA1c 3 years ago	1008.000	77/26	0.048
LDL currently	997.500	77/26	0.979
LDL 3 years ago	1052.500	77/26	0.696
HDL currently	1137.500	77/26	0.300
HDL 3 years ago	1167.000	77/26	0.207
TG currently	948.500	77/26	0.690
TG 3 years ago	873.000	77/26	0.331
Cholesterol currently	992.500	77/26	0.949
Cholesterol 3 years ago	943.000	77/26	0.660
NHDL currently	64.000	23/6	0.813
ADL	460.000	77/26	*p* < 0.001
IADL	461.500	77/26	*p* < 0.001
GDS	1636.500	77/26	*p* < 0.001
Grip strength right hand: attempt 1	388.000	77/26	*p* < 0.001
Grip strength right hand: attempt 2	365.000	77/26	*p* < 0.001
Grip strength left hand: attempt 1	254.000	77/26	*p* < 0.001
Grip strength left hand: attempt 2	223.500	77/26	*p* < 0.001

## Data Availability

The datasets generated and analyzed during the current study are available from the corresponding author on reasonable request.

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
