# Peer review of "Implications of the Metabolic Control of Diabetes in Patients with Frailty Syndrome"

_ijerph, 2022, doi:10.3390/ijerph191610327_

Round 1
Reviewer 1 Report
Overall, the paper is very interesting, the population of the study is sufficiently large and well balance among male and female participants. However, authors should better discuss the applicability of their results/findings, What is the next step for the people diagnosed with frailty syndrome? Is something being done regarding their eating habits and exercise? I strongly believe these questions should be answered within the paper. Otherwise why would any physicians apply these results?
Overall comments:
Moderate English changes should be done throughout the whole manuscript.
Minor misprints lines 34, 115, 117, 181, 238, 247, Table 2.
There is an overused of the phrase ‘frailty syndrome’ in the abstract.
There is an invalid change of format from lines 129-152.
Accurate subsection should be identified in the Study design section for each scale.
Major redaction error in lines 238-239.
I advise the authors to add some figures illustrating the statistical results of section 2.5.
Lines 288-290 does not make any sense.
Line 296, is Table 3?
Column ‘Statistics’ in Table 3 is confusing; Authors should better explain these values in the caption.
Author Response
Thank you very much for all your opinions and suggestions
Reviewer 1
Overall, the paper is very interesting, the population of the study is sufficiently large and well balance among male and female participants. However, authors should better discuss the applicability of their results/findings, What is the next step for the people diagnosed with frailty syndrome? Is something being done regarding their eating habits and exercise? I strongly believe these questions should be answered within the paper. Otherwise why would any physicians apply these results?
Corrected in introduction
Our study is comprehensive, and multi-stage, with the use of geriatric assessment elements, many research tools, as well as physical and biochemical tests. It is also retrospective and comparative, which is not found in other studies. Such a multi-faceted approach broadens the image and knowledge of the functioning of patients with diabetes and frailty. Each patient diagnosed with frailty was subjected to further detailed examinations. Each patient was provided education on further dietary management and physical exercises individually. The latest knowledge in this field guided the doctor.
Overall comments:
- Moderate English changes should be done throughout the whole manuscript.
Corrected
- Minor misprints lines 34, 115, 117, 181, 238, 247, Table 2.
Corrected
- There is an overused of the phrase ‘frailty syndrome’ in the abstract.
Corrected
- There is an invalid change of format from lines 129-152.
Corrected
- An accurate subsection should be identified in the Study design section for each scale.
Corrected
- Major redaction error in lines 238-239.
Corrected
- I advise the authors to add some figures illustrating the statistical results of section 2.5.
Added
- Lines 288-290 do not make any sense.
Corrected
- Line 296, is Table 3?
Yes, corrected.
- Column ‘Statistics’ in Table 3 is confusing; the Authors should better explain these values in the caption.
Corrected

Reviewer 2 Report
The study is interesting and important for the readership for IJERPH. However, it has some minor corrections needs to be considered before considering this work for publication as detailed below
1.If possible, emphasis on mechanisms of lowering glucose and cognitive decline on frailty syndrome. Quote some mechanisms related papers either preclinical or clinical studies.
2.Describe how your work is difference from other work and emphasis on novelty findings. Make a change in the introduction part.
3.Please correct the spelling mistake in Table 2 as ‘Maximum”
Author Response
Reviewer 2
The study is interesting and important for the readership of IJERPH. However, it has some minor corrections that need to be considered before considering this work for publication as detailed below
- If possible, emphasis on mechanisms of lowering glucose and cognitive decline on frailty syndrome. Quote some mechanisms-related papers either preclinical or clinical studies.
We think that such a detailed description of the mechanisms of lowering glucose levels in people with frailty in this article would not be recommended. Readers of typically clinical scientific journals will probably find such a description in them.
- Describe how your work is different from other work and emphasize novelty findings. Make a change in the introduction part.
Added to introduction
Our study is comprehensive, and multi-stage, with the use of geriatric assessment elements, many research tools, as well as physical and biochemical tests. It is also retrospective and comparative, which is not found in other studies. Such a multi-faceted approach broadens the image and knowledge of the functioning of patients with diabetes and frailty. Each patient diagnosed with frailty was subjected to further detailed examinations. Each patient was provided education on further dietary management and physical exercises individually. The latest knowledge in this field guided the doctor.
3. Please correct the spelling mistake in Table 2 as ‘Maximum”
Corrected
